# The Association between Seafood Intake and Fecundability: Analysis from Two Prospective Studies

**DOI:** 10.3390/nu12082276

**Published:** 2020-07-29

**Authors:** Lauren A. Wise, Sydney K. Willis, Ellen M. Mikkelsen, Amelia K. Wesselink, Henrik Toft Sørensen, Kenneth J. Rothman, Katherine L. Tucker, Ellen Trolle, Marco Vinceti, Elizabeth E. Hatch

**Affiliations:** 1Department of Epidemiology, Boston University School of Public Health, Boston, MA 02118, USA; siwillis@bu.edu (S.K.W.); akw23@bu.edu (A.K.W.); hts@clin.au.dk (H.T.S.); kenneth.rothman@gmail.com (K.J.R.); marco.vinceti@unimore.it (M.V.); eehatch@bu.edu (E.E.H.); 2Department of Clinical Epidemiology, Aarhus University Hospital, Nordre Ringgade 1, Aarhus C, 8000 Aarhus, Denmark; em@clin.au.dk; 3RTI Health Solutions, Research Triangle Park, NC 27709, USA; 4Department of Biomedical and Nutritional Sciences, College of Health Sciences, University of Massachusetts Lowell, Pawtucket St, Lowell, MA 01854, USA; Katherine_Tucker@uml.edu; 5National Food Institute, Technical University of Denmark, 2800 Kgs Lyngby, Denmark; eltr@food.dtu.dk; 6Department of Biomedical, Metabolic and Neural Sciences, University of Modena and Reggio Emilia, 41121 Modena, Italy

**Keywords:** seafood, fish, fatty acids, fertility, internet, epidemiology, prospective studies

## Abstract

Background: Seafood is an important source of omega-3 fatty acids, which have been associated with improved oocyte quality and embryo morphology in some studies. However, seafood is also a source of persistent organic pollutants and heavy metals, which may adversely affect fecundity. Previous studies of seafood intake and fecundity have generated inconsistent results. Methods: In two prospective cohort studies of 7836 female pregnancy planners from Denmark (Snart Foraeldre, *n* = 2709) and North America (PRESTO, *n* = 5127), we evaluated the association of dietary intake of total seafood and marine-sourced long-chain omega-3 fatty acids (eicosapentaenoic acid, docosahexaenoic acid, and docosapentaenoic acid) with fecundability. Participants completed a baseline questionnaire on sociodemographics, behavioral factors, anthropometrics, and medical history, and a food frequency questionnaire. Pregnancy status was updated bimonthly for up to 12 months or until reported conception. We estimated fecundability ratios (FRs) and 95% confidence intervals (CIs) using proportional probabilities regression models, adjusted for energy intake and other potential confounders. We restricted analyses to women with ≤6 menstrual cycles of attempt time at enrollment. Results: Intake of total seafood or marine-sourced long-chain omega-3 fatty acids was not appreciably associated with fecundability in either cohort (≥200 vs. <50 g/week total seafood: FR = 0.94, 95% CI: 0.79–1.10 in Snart Foraeldre; FR = 1.00, 95% CI: 0.90–1.13 in PRESTO; marine fatty acids: ≥90th vs. <25th percentile: FR = 1.00, 95% CI: 0.85–1.18 in Snart Foraeldre; FR = 0.97, 95% CI: 0.86–1.09 in PRESTO). In PRESTO, where we collected additional data on seafood preparation, we observed an inverse association between fecundability and fried shellfish (≥10 g/week vs. none: FR = 0.77, 95% CI: 0.61–0.98), but not unfried shellfish (≥20 g/week vs. none: FR = 0.98, 95% CI: 0.89–1.07); in Snart Foraeldre, there was no association with total shellfish intake. Conclusions: We found little association between seafood intake and fecundability overall, but greater intake of fried shellfish was associated with reduced fecundability among North American participants.

## 1. Introduction

Roughly 10–15% of couples experience infertility, which is defined clinically as a lack of conception after 12 months of unprotected intercourse [1]. Seafood is an important source of omega-3 fatty acids, which are important for steroidogenesis [2] and have anti-inflammatory effects [3,4]. Greater intake of omega-3 fatty acids via fish or dietary supplementation has been associated with enhanced oocyte maturation and development [2], reduced probability of anovulation and higher progesterone concentrations [5], and improved embryo morphology [6,7,8] in animal and human studies. Moreover, some [9,10] but not all [11,12,13] prospective cohort studies of omega-3 fatty acids (measured via diet or blood) have shown positive effects on fecundability. At the same time, seafood also contains persistent organic pollutants and trace elements (e.g., heavy metals) [14], which may have adverse effects on reproductive hormones and fecundity [15,16,17,18,19].

There have been three retrospective cohort studies [20,21,22] and one prospective cohort study [23] of seafood intake and time-to-pregnancy (TTP), and one prospective cohort study of infertility treatment outcomes [24], with inconsistent results. In a retrospective study of 3421 pregnant French women, greater shellfish intake was associated with longer TTP (≥2 servings/week vs. <2 servings/month: fecundability odds ratio = 0.71); there was little association with total fish consumption [20]. Among 1234 female participants from the New York State Angler Cohort, greater fish intake from the Great Lakes (i.e., waters with suspected polychlorinated biphenyl contamination) was associated with reduced fecundability (3–6 years of consumption vs. none: fecundability ratio (FR) = 0.75) [21]. In a Swedish retrospective cohort study of fishermen’s sisters, greater consumption of locally-caught fatty fish was associated with shorter TTP (greater fecundability) among women residing on the east coast, an area with high levels of persistent organochlorine chemicals [22]. In that study, FRs comparing medium (1–1.5 meals/month) and high (≥2 meals/month) consumption of fatty fish vs. low consumption (≤0.5 meals/month) were 1.16 and 1.27, respectively [22]. In a prospective preconception cohort study of 501 couples from Michigan and Texas, seafood intake was associated with shorter TTP (≥8 vs. ≤1 servings/month: FR = 1.44) [23]. Finally, in a 2018 study of 351 women receiving infertility treatment, fish consumption was not appreciably associated with the probability of clinical pregnancy (quartile 4 vs. 1: 51.2% vs. 47.4%), but was positively associated with a higher probability of live birth (quartile 4 vs. 1: 47.7% vs. 34.2%) [24]. Thus, existing studies of seafood intake and TTP have generally been small, retrospective or based in infertile populations, or have been limited in their exposure assessment.

We examined prospectively the association between seafood consumption and fecundability in two preconception cohorts of women residing in Denmark and North America. Specifically, we evaluated fecundability in relation to intake of total seafood and marine-sourced long-chain omega-3 fatty acids (i.e., eicosapentaenoic acid, docosahexaenoic acid, and docosapentaenoic acid). Finally, in our North American cohort, where we collected additional data on food preparation, we further evaluated intake of fried vs. unfried seafood and fecundability.

## 2. Methods

### 2.1. Study Population

The present analysis is based on data from two web-based preconception cohort studies. Snart Foraeldre (SF, “Soon Parents”) is a prospective cohort study of pregnancy planners in Denmark. SF is an expansion of the Snart Gravid (“Soon Pregnant”) study [25,26]. SF recruitment began in 2011 with advertisements placed on Danish health-related websites. Beginning in 2018, potential participants were invited via E-box, a Danish communication platform for Danish authorities and citizens. Enrollment and data collection were carried out using online self-administered questionnaires. Beginning in January 2013, ten days after cohort entry, participants were asked to complete a food frequency questionnaire (FFQ) specifically designed for and validated within this study population [27]. Eligible women were aged 18–45 years, residents of Denmark, actively trying to conceive, and not receiving fertility treatment. Baseline questionnaires included information on socio-demographics, behavioral factors, anthropometrics, and reproductive and medical history. To update pregnancy status and changes in exposures, self-administered online follow-up questionnaires were completed every 8 weeks for 12 months or until a reported conception. The SF protocol is registered at Aarhus University in compliance with Danish law on data protection and approved by the Institutional Review Board at Boston University Medical Campus. All participants provided online informed consent.

Pregnancy Study Online (PRESTO) is a North American prospective cohort study that was modeled after SF and initiated in 2013 [28]. Eligible women were aged 21–45 years, residents of Canada or the U.S., actively trying to conceive, and not receiving fertility treatment. Women completed baseline and follow-up questionnaires similar to those administered in SF. Ten days after enrollment, PRESTO participants were invited to complete the web-based version of the National Cancer Institute’s Diet History Questionnaire (DHQ) II [29], a FFQ validated in a U.S. population. PRESTO was approved by the Institutional Review Board of the Boston University Medical Campus. All participants provided online informed consent (IRB protocol number: #H-31848). Both studies were designed according to the STROBE Statement (https://strobe-statement.org/). Details regarding exclusions for the present analysis are provided in Figure 1.

### 2.2. Assessment of Seafood and Marine Fatty Acid Intake

We estimated intake of seafood and fatty acids using the nutrient composition of all food items in the FFQ. The FFQ was validated in each population against 24-h recalls or food records [27,29,30]. In SF, we asked about intake of fish and seafood dishes (lean fish such as cod, pollock, plaice; oily fish such as salmon, herring, or mackerel; seafood (e.g., shrimp, crayfish, lobster tails); other fish dishes, such as fish cakes or fish lasagna; and sushi), fish products on bread (pickled or smoked fish (e.g., herring, salmon or mackerel in tomato sauce); fried fish (e.g., fish filet or fish ball); shrimp, mussels, crabs and the like; tuna (e.g., tuna in water, oil, or tomato); mayonnaise salad (e.g., tuna, mackerel or shrimp salad); or cod roe), food pie with or without meat or fish, other soups with or without meat or fish, and green salad with fish. We also extracted seafood intake from mixed recipes. In PRESTO, we asked about intake of fried shellfish; shellfish that were not fried; salmon, fresh tuna or trout; canned tuna (including in salads, sandwiches, or casseroles); fish sticks or other fried fish (not including shellfish); and other fish that was not fried (not including shellfish). We also extracted information on seafood intake from mixed recipes. All questions in both cohorts were asked with respect to the previous 12 months (Appendix A).

We calculated total seafood intake by summing all servings of fish and other seafood from individual foods and mixed recipes on the FFQ. In SF, we obtained information about the fatty acid content of specific foods from the Danish Food Composition Databank revision 7. National Food Institute, Technical University of Denmark, 2008 (www.foodcomp.dk). In PRESTO, we used the National Cancer Institute’s DIET*CALC software (version 1.5.0, Rockville, MD, USA) [31] to estimate fatty acid consumption. In both cohorts, we calculated total marine-sourced long-chain omega-3 fatty acids (“marine fatty acids”) by summing intakes of eicosapentaenoic acid (EPA), docosahexaenoic acid (DHA), and docosapentaenoic acid (DPA). The deattenuated correlation coefficient for fish intake was 0.75 comparing the SF FFQ to a 4-day food diary in the SF validation study [27], and 0.53 comparing PRESTO’s DHQ II FFQ to four 24-h recalls in the National Cancer Institute’s validation study [29].

### 2.3. Assessment of Time to Pregnancy

On the baseline questionnaire, women reported the number of menstrual cycles in which they had been trying to conceive. At baseline and during follow-up, we collected data on date of last menstrual period (LMP) and cycle regularity (“Has your menstrual period been regular in a way that you could usually predict about when the next period would start?”). Women with regular cycles were asked about their typical cycle length; women with irregular cycles had their cycle length estimated based on the number of menses in a year, the number of days until their next period was expected, and LMP dates at baseline and follow-up.

On follow-up questionnaires administered every 8 weeks, women reported if they were currently pregnant, if they had initiated fertility treatment, and if they had experienced any intervening pregnancy losses since their previous questionnaire. Women who reported conception were asked for details about how their conceptions were confirmed (e.g., home pregnancy test and/or blood test in doctor’s office). Over 96% of participants reported using home pregnancy tests to confirm pregnancy [28]. Women who did not report conception were asked if they were still trying to conceive. Among PRESTO participants who were lost to follow-up, we sought additional outcome information by contacting participants directly via phone or email, searching for baby registries and birth announcements online, and by linking with birth registries in selected states (CA, FL, MA, MI, OH, PA, and TX). When we identified pregnancies without direct participant contact, we assumed that this was the woman’s first pregnancy since she enrolled in the study. We estimated TTP in discrete menstrual cycles using the following formula: [(reported menstrual cycles of pregnancy attempt time at baseline) + [(LMP date from most recent follow-up questionnaire − date of baseline questionnaire)/cycle length] + 1].

### 2.4. Assessment of Covariates

On the baseline questionnaire, participants reported information on potential confounders, including age, height, weight, physical activity, smoking, alcohol consumption, education, household income, marital status, race/ethnicity (PRESTO only), last method of contraception, parity, and the use of dietary supplements (including fish oil supplements). We calculated body mass index (BMI) as weight (kg) divided by height squared (m^2^). In SF, total weekly metabolic equivalents (METs) were calculated using the International Physical Activity Questionnaire short-form by summing the MET-hours from walking, moderate physical activity, and vigorous physical activity (hours/week × 3.3 METs, 4 METs, and 8 METs, respectively) [32]. In PRESTO, total MET-hours per week were calculated by multiplying the average number of hours per week spent engaging in various activities by metabolic equivalents estimated from the Compendium of Physical Activities [33,34]. Based on the dietary questionnaires, we estimated total energy intake and assessed overall diet quality (measured via the Nutrient Rich Diet Score in SF and the Healthy Eating Index 2010 in PRESTO) [35,36]. The list of potential confounders examined in the two cohorts was identical, except for race/ethnicity (ascertained in PRESTO only).

### 2.5. Data Analysis

We performed parallel analyses across the two cohorts. Seafood variables were categorized according to their frequency distributions in the cohort. We analyzed omega-3 fatty acids in five categories, with cut points at the 25th, 50th, 75th, and 90th percentiles, both individually and grouped as marine fatty acids (EPA + DPA + DHA). Participants contributed menstrual cycles to the analysis until they reported a conception or one of the following censoring events: cessation of pregnancy attempts, initiation of fertility treatment, study withdrawal, loss to follow-up, or 12 cycles, whichever came first. To account for variation in pregnancy attempt time at cohort entry (range: 0–6 cycles) and to reduce bias from left truncation [37,38], we based risk sets on observed cycles at risk only using the Andersen–Gill data structure [39]. We used life-table methods to calculate the percentage of couples that conceived during 12 cycles of follow-up, accounting for censoring [40]. We used proportional probabilities regression models [41,42] to estimate fecundability ratios (FRs), defined as the ratio of the cycle-specific probability of conception among exposed compared with unexposed women. This model controls for the expected decline in cohort fecundability over time by adjusting for binary indicators of cycle number at risk.

Our selection of control variables was guided by the literature and by causal diagrams. We considered known or suspected determinants of subfertility that were associated with total seafood intake. Our final models adjusted for age (<25, 25–29, 30–34, 35–39, or ≥40 years), cigarette smoking (never, former, current occasional, or current regular), alcohol use (drinks/week), BMI (<20, 20–24, 25–29, 30–34, or ≥35 kg/m^2^), intake of sugar-sweetened beverages (drinks/week), physical activity (<10, 10–19, 20–39, or ≥40 MET-hours/week), last contraceptive method used (hormonal, barrier, or natural methods), intercourse frequency (<1, 1, 2–3, or ≥4 times per week), parity history (0 or ≥1 births), use of method(s) to improve pregnancy chances (yes/no), daily use of prenatal or multivitamins (yes/no), fish oil supplement use (yes/no), and marital status (married/living as married vs. not). In addition, PRESTO models controlled for education (≤high school, some college, college degree, or graduate school), household income (<50,000, 50,000–99,999, 100,000–149,999, or ≥150,000 United States Dollars (USD)/year), race/ethnicity (non-Hispanic White: yes vs. no), and Healthy Eating Index 2010 score (continuous) while SF models controlled for educational training duration (≤12, 13–15, 16, or ≥17 years), household income (<25,000, 25,000–39,999, 40,000–64,999, or ≥65,000 Danish Kroners (DKK)/month), and Nutrient Rich Dietary Score (continuous). For foods, we adjusted for total energy intake by including a continuous term for energy intake in the regression model, except for fatty acids, where we used the nutrient residual method [43]. Additional sensitivity analyses were restricted to non-users of fish oil supplements to better isolate the effect of seafood intake on fecundability. Analyses were also performed with and without adjustment for intercourse frequency, which we considered a potential confounder, but which others have treated as a potential mediator of the seafood-fecundability association [23].

We performed secondary analyses that were stratified by pregnancy attempt time at cohort entry (<3 vs. 3–6 cycles) to assess the degree to which reverse causation explained our associations (e.g., if subfertility caused changes in seafood intake). We also stratified models by age, parity, and BMI, given their strong relationship to fecundability and the fact that some chemicals in seafood (e.g., polychlorinated biphenyls) are strongly correlated with adiposity (via storage in body fat) [44] and parity (via excretion through pregnancy and lactation) [45]. Out of concern that controlling for parity might introduce bias due to overcontrol for underlying fecundity [46,47], models were fit with and without adjustment for parity.

We used multiple imputation to impute missing covariate and outcome data [48,49]. We assigned women with no follow-up data (SF: *n* = 177; PRESTO: *n* = 144) one cycle of follow-up and imputed their pregnancy status at the end of that cycle. Covariate missingness in SF ranged from 0% (age, fish intake, and energy intake) to 10% (household income). In PRESTO, covariate missingness ranged from 0% (age, parity, marital status, education, fish intake, and energy intake) to 4% (household income). Within each cohort, we created five imputed datasets and statistically combined coefficients and standard errors across the five datasets using PROC MIANALYZE. We carried out all analyses using SAS version 9.4 (Cary, NC, USA) [50].

## 3. Results

During 2013–2019, 2709 SF participants contributed 1818 pregnancies and 9609 menstrual cycles of pregnancy attempt time and 5127 PRESTO participants contributed 3267 pregnancies and 21,076 menstrual cycles of pregnancy attempt time. Using life-table methods, the percentage of couples that conceived during 12 cycles of follow-up after accounting for censoring was 82% in SF and 74% in PRESTO. During follow-up in SF, 3% stopped trying to conceive, 6% initiated fertility treatment, 15% were lost to follow-up, 8% were censored at 12 cycles, and 1% were still actively participating in the study. During follow-up in PRESTO, 3% stopped trying to conceive, 11% initiated fertility treatment, 7% were lost to follow-up, 10% were censored at 12 cycles, and 4% were still actively participating in the study. The distributions of total seafood intake differed across the two cohorts, with SF participants reporting substantially higher intake of all types of seafood (SF: median = 142.3, interquartile range (IQR) = 88.8–211 g/week; PRESTO: median = 55.6, IQR = 19.8–115 g/week).

Table 1 displays baseline characteristics of study participants by categories of total seafood intake. In general, greater seafood intake was associated with higher socioeconomic status and healthier diet and behaviors. Specifically, total seafood intake was positively associated with physical activity, intercourse frequency, daily multivitamin use, fish oil supplementation, indices of healthy diet (Nutrient Rich Diet Score and Healthy Eating Index score), education, and household income, and inversely associated with BMI, parity, and cigarette smoking in both cohorts. While total seafood intake was positively associated with alcohol intake in PRESTO, there was no appreciable association in SF. Finally, in PRESTO, total seafood intake was lower among those who identified as non-Hispanic White as compared with other racial/ethnic groups.

Total seafood intake was not appreciably associated with fecundability in either cohort (≥200 vs. <50 g/week total seafood: FR = 0.94, 95% CI: 0.79–1.10 in SF; FR = 1.00, 95% CI: 0.90–1.13 in PRESTO) (Table 2). Likewise, greater intake of shellfish overall was not materially associated with fecundability in either cohort. However, in PRESTO, where we collected additional data on seafood preparation, we observed an inverse association between fried shellfish (≥10 g/week vs. none: FR = 0.77, 95% CI: 0.61–0.98) and fecundability, but there was no evidence of a dose-response relation. We observed no association between unfried shellfish and fecundability (≥20 g/week vs. none: FR = 0.98, 95% CI: 0.89–1.07). There was minimal evidence of confounding when comparing unadjusted and adjusted associations.

Dietary intake of individual marine-sourced long-chain omega-3 fatty acids or their summation showed little association with fecundability in either cohort (Table 3).

Restricted cubic spline curves were consistent with the categorical results (Figure 2).

Associations were generally consistent across strata of age and BMI (Appendix A). Among PRESTO participants with shorter attempt times at cohort entry, among whom diet was less likely to have changed in response to subfertility, the FRs for fried shellfish persisted (≥10 g/week vs. none: FR = 0.74, 95% CI: 0.56–0.97) (Appendix A). All other results were generally similar across strata. Stratifying by parity, or omitting parity altogether from the multivariable models, resulted in little change in the FRs (data not shown).

Use of fish oil supplements was relatively common in both cohorts (18–20%) but was not materially associated with fecundability in either cohort (Table 2). We observed similar associations for total seafood intake and intake of marine fatty acids with fecundability among non-users of fish oil supplements in PRESTO (Appendix A). Results were similar with and without control for frequency of intercourse (data not shown).

## 4. Discussion

We found little evidence of an association between intake of total seafood or marine fatty acids and fecundability. Results did not vary appreciably by BMI, age at cohort entry, or parity. Controlling for several measures of socioeconomic status (e.g., education, household income, and marital status) or indices of healthy diet did not materially change these associations. In PRESTO, where we had additional data on fish preparation, there was some suggestion that greater intake of fried shellfish was associated with reduced fecundability, but there was no dose-response relation.

Previous studies of the association between seafood intake and fecundability have produced conflicting results [20,21,23]. Our finding regarding fried shellfish intake and reduced fecundability is consistent with a retrospective TTP study of 3421 French pregnant women showing that greater shellfish intake was associated with longer TTP (≥2 servings/week vs. <2 servings/month: fecundability odds ratio = 0.71) [20]. However, French populations are more likely to eat pan-fried/sauteed or baked shellfish [51] than deep-fried shellfish, which is more commonly consumed by North Americans. Our results for total seafood intake did not agree with the positive findings observed for fish intake and live birth rates in the EARTH study [24], but both studies observed nearly null associations with fish oil supplementation. Though we did not confirm the overall positive results from the LIFE study regarding total seafood intake and fecundability [23], we did observe a positive association between greater seafood intake and intercourse frequency. However, additional control for intercourse frequency, a potential confounder, made little difference in our associations between seafood intake and fecundability. The inconsistent results across studies could be explained by possible opposing effects of seafood intake on fecundability, including potentially beneficial (e.g., omega-3 fatty acids) and harmful (e.g., chemical toxicants) effects. Ideally, studies would measure both fatty acids and toxicants in the same individuals, and evaluate these exposures simultaneously to disentangle their independent effects on fecundability.

The results for fried shellfish are not wholly surprising. Other studies have found associations between fried seafood (shellfish or fish) intake and adverse health outcomes (e.g., diabetes [52], cancer [53], all-cause and cardiovascular mortality [54,55]), even when overall associations with total seafood intake showed no association [52,54] or non-fried seafood showed beneficial effects [56]. There is some evidence that the preparation of seafood influences the extent to which contaminants are retained in the flesh of the seafood [57,58,59,60,61]. These studies suggest that frying seafood, particularly fatty seafood harvested from contaminated waters, is potentially harmful because it can seal in organochlorines and other chemical pollutants. There is also evidence that frying foods produces new contaminants (heterocyclic amines [62,63], polycyclic aromatic hydrocarbons [64] and, to a smaller extent, acrylamide [65]), and those who eat more fried seafood tend to eat other fried foods. If the shellfish consumed by study participants contained more contaminants on average than other fish, this could explain the stronger results for fried shellfish than fried seafood more generally [14].

Study strengths include the enrollment of participants during the preconception period, with more than 65% enrolled within three cycles of discontinuing contraception. The assessment of diet and other covariates before pregnancy helped to avoid recall bias. Loss to follow-up was lower than 20% in both cohorts during the study period. There were no appreciable differences in the proportion lost-to-follow-up in the extreme quartiles of total seafood intake. Data were collected on several potential confounders. FFQs were validated in each respective population, showing relatively high agreement of the FFQ when compared with 24-h recalls and/or food diaries.

Several potential study weaknesses also must be considered. Although the FFQ is the most practical and efficient method for measuring long-term diet in large epidemiologic studies [43,66,67], some misclassification of diet is anticipated. Cohort-specific dietary databases were used in the two studies, but many nutrients were based on laboratory analysis with extrapolation from similar foods. The FFQ for SF contained many more seafood items (*n* = 14) than PRESTO (*n* = 6), including mixed recipes (e.g., fish stew), and intake of seafood was much greater in Denmark than in North America, making comparisons across cohorts difficult. Misclassification may have also arisen from differences in the types of seafood, its fat content, preparation of the seafood (deep-fried, pan-fried/sauteed, grilled, or baked), and contamination of waters from which the seafood was harvested. With prospective assessment of dietary intake relative to pregnancy, we would expect misclassification to be non-differential, which could have unpredictable effects on the inner categories of exposure but would generally bias extreme exposure categories towards the null. Thus, non-differential misclassification could be an explanation of our null findings for most seafood variables, though we note that previous studies with positive results have also relied on FFQs.

Residual confounding may have also influenced our results. Consumption of foods containing omega-3 fatty acids (e.g., unfried wild-caught fish) is associated with more healthful lifestyle practices. We cannot rule out confounding by healthful lifestyle factors not captured by measured variables such as SES, physical activity, or energy intake. For example, the inverse association for fried shellfish could be explained by residual confounding if the healthiest women with high fecundability are more likely to be classified in the lowest categories of exposure. Another limitation is the lack of adjustment for male partners’ dietary intake, which is likely correlated with female partners’ intake. Male seafood intake has been shown to influence reproduction in some studies (e.g., DHA deficiency is associated with poor semen quality) [68,69,70]. Finally, there could be other factors that explain the inconsistent results across cohorts, particularly for fried shellfish and seafood high in omega-3 fatty acids (e.g., changes in contaminant levels over calendar time [71]; differences in pollution of the fish supply in North America [72] vs. Denmark [73]).

Given the lack of effect measure modification by selected covariates (e.g., age, BMI, and parity), our results may be generalizable to a broader population of reproductive-aged women. Internet-based recruitment should not influence the validity of the measures of association unless the association between diet and fertility differs markedly between internet users and nonusers, which seems unlikely [74]. The same logic can be applied regarding differences between planners and non-planners of pregnancy. Our study [75] and others [76,77] have shown that even when enrollment at study entry is related to characteristics such as age, cigarette smoking, or parity measures of association are not appreciably biased due to self-selection. Other potential biases, such as selection bias stemming from length-biased recruitment of couples with longer TTP and reverse causation resulting from subfertility-related changes in diet, were assessed empirically by stratifying on attempt time at cohort entry. Selected results (e.g., fried shellfish) were slightly stronger among women with shorter attempt times at cohort entry.

In conclusion, there was little evidence of an overall association between intake of total seafood or marine fatty acids and fecundability in two prospective cohort studies of North American and Danish pregnancy planners. We observed, however, an inverse association between fecundability and intake of fried shellfish, but not unfried shellfish, among North American women.

## Figures and Tables

**Figure 1 nutrients-12-02276-f001:**
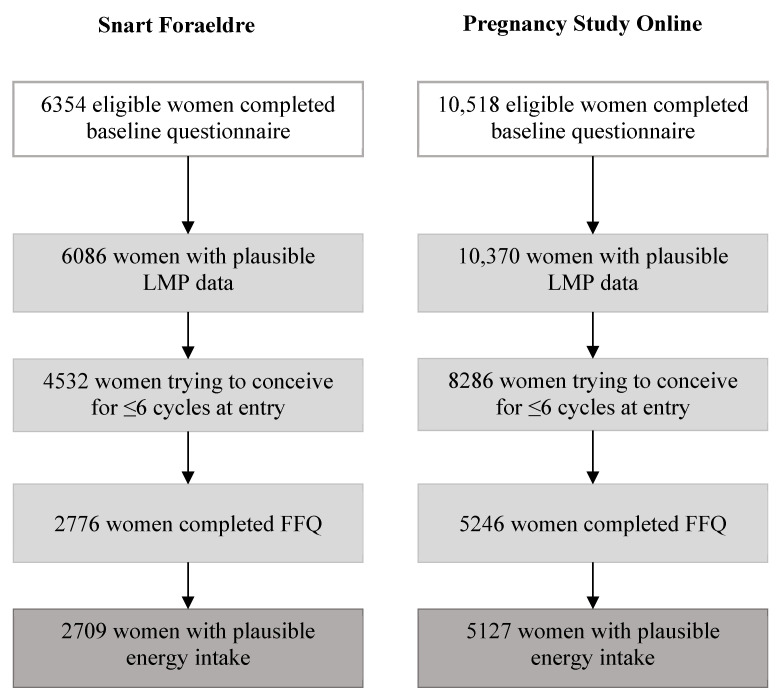
Flowchart of participant exclusions, 2013–2018 (Snart Foraeldre) and 2013–2019 (PRESTO).

**Figure 2 nutrients-12-02276-f002:**
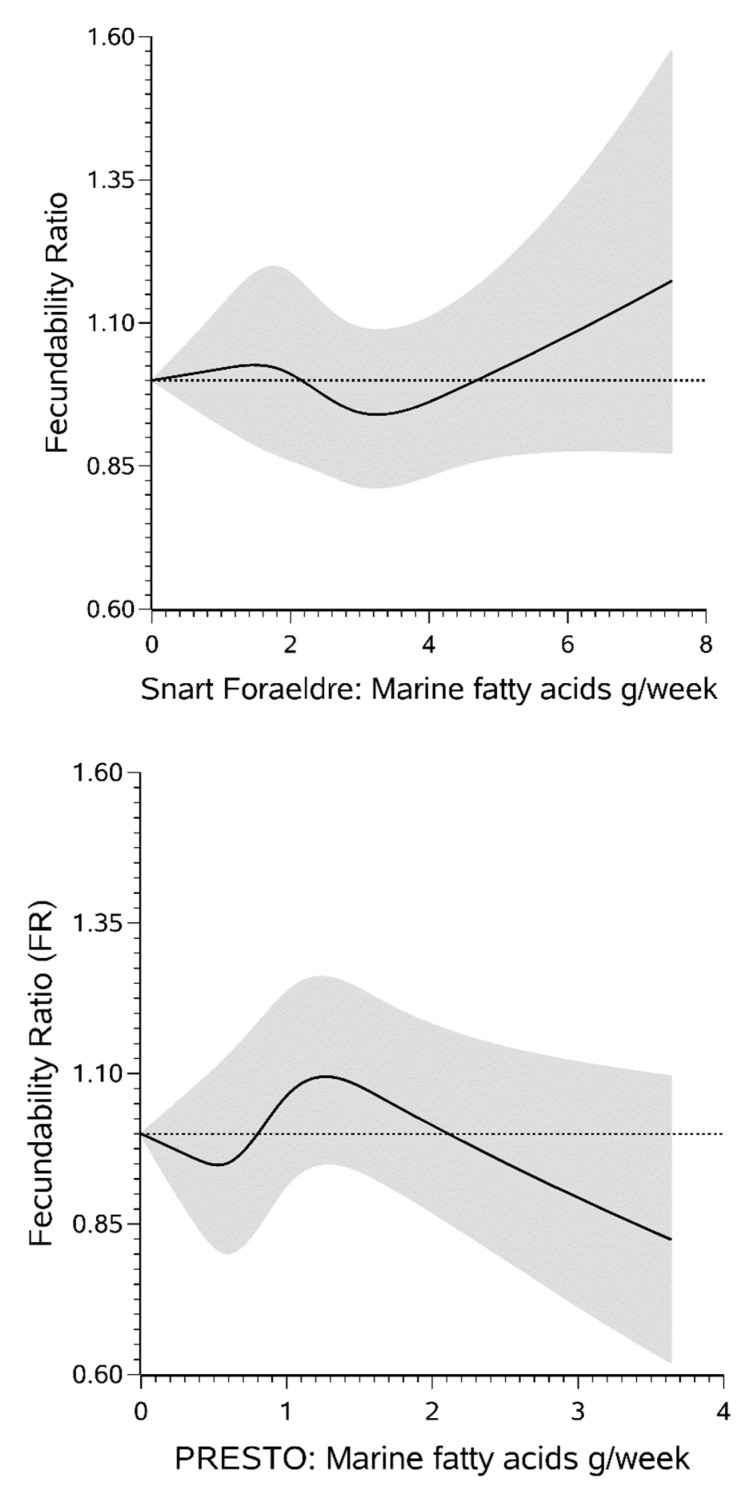
Association between marine fatty acid and fecundability among 2709 female Snart Foraeldre participants (**top**) and 5127 female Pregnancy Study Online (PRESTO) participants (**bottom**), fitted by restricted cubic splines. The reference level for the fecundability ratio is the lowest value in the data in each cohort. The splines are trimmed at the 99th percentile and have 4 knot points at the 25th, 50th, 75th and 90th percentiles. For Snart Foraeldre, knot points are at 1.00, 1.84, 2.90, and 4.30 and for PRESTO, knot points are at 0.36, 0.59, 1.00, and 1.64. Both splines are adjusted for age, BMI, physical activity, smoking status, alcohol intake, last form of contraception, intercourse frequency, parity, use of methods to improve pregnancy chances, daily multivitamin use, use of fish oil supplements, education, income, marital status, race/ethnicity (PRESTO only), healthy diet indices, and sugar-sweetened beverage intake.

**Table 1 nutrients-12-02276-t001:** Baseline characteristics of participants by dietary intake of seafood, 2013–2018 (Snart Foraeldre) and 2013–2019 (PRESTO) ^a^.

	Total Seafood Intake (g/week)							
	Snart Foraeldre (*n* = 2709)				PRESTO (*n* = 5127)			
	<50	50–99	100–199	≥200	<50	50–99	100–199	≥200
Number of women (*n*)	316	509	1117	767	2343	1257	1020	507
Age, mean (years)	28.4	28.8	28.9	29.4	29.6	30.3	30.5	30.7
Body mass index, mean (kg/m^2^)	25.0	24.0	24.0	23.8	27.9	26.8	26.5	26.6
Energy, mean (kcal/day)	1675	1714	1824	2097	1466	1572	1718	1842
Current smoker (%)	6.8	8.3	5.0	3.6	6.1	3.1	3.9	3.0
Parous (%)	32.0	32.0	33.2	31.1	34.8	30.8	28.2	21.3
Alcohol intake, mean (drinks/week)	2.6	2.8	2.7	2.8	2.7	3.5	3.6	4.0
Physical activity, mean (METs/week)	64.9	62.8	59.7	71.9	31.9	34.5	38.3	43.6
Last contraceptive: hormonal (%)	57.6	60.9	59.5	54.1	39.2	38.4	38.1	37.1
Intercourse (frequency/week) (%)								
≤1	44.1	41.7	39.2	36.8	40.3	42.5	41.6	36.2
2–3	43.6	46.0	47.2	46.9	45.4	43.0	45.3	48.7
≥4	12.3	12.3	13.6	16.3	14.3	14.5	13.1	15.1
Daily multivitamin use (%)	65.3	66.7	70.0	75.5	81.4	84.7	86.6	87.4
Fish oil supplement intake (%)	17.6	19.7	17.9	21.2	16.4	21.2	20.3	24.8
Healthy diet indices, NRDS/HEI (mean)	979	1015	1033	1062	63.1	67.1	69.0	71.0
Sugar-sweetened beverages/week (mean)	1.3	0.9	0.8	0.6	2.8	2.2	2.1	1.9
White/non-Hispanic (%)	100	100	100	100	88.4	86.5	85.8	82.9
Education, years (%)								
≤12	10.0	3.4	4.6	3.6	4.2	2.1	1.7	2.6
13–15	21.4	15.3	12.9	11.9	22.7	15.9	15.9	13.4
16	36.5	38.4	37.7	37.8	33.7	36.8	35.4	31.2
≥17	32.0	42.9	44.7	46.7	39.3	45.2	47.0	52.7
Income, DKK/month or USD/year (%)								
<25,000/<50,000	16.1	12.8	12.1	12.5	18.8	16.1	13.3	15.1
25,000–39,999/50,000–99,999	24.3	22.2	21.2	22.8	42.2	37.6	36.7	32.7
40,000–64,999/100,000–149,999	39.6	45.6	43.3	41.8	24.4	31.0	28.8	24.4
≥65,000/≥150,000	20.0	19.3	23.4	23.0	14.6	15.4	21.3	27.8

Kcal = kilocalories, MET = metabolic equivalent of task, NRDS = Nutrient Rich Diet Score, HEI = Healthy Eating Index 2010, DKK = Danish Kroners, USD = United States dollars. ^a^ All variables, except for age, are standardized to the age distribution of the study population at baseline.

**Table 2 nutrients-12-02276-t002:** Association between seafood intake and fecundability, Snart Foraeldre and PRESTO cohorts.

	Snart Foraeldre						PRESTO					
	No. of Pregnancies	No. of Cycles	FR ^a^	95% CI ^a^	FR ^b^	95% CI ^b^	No. of Pregnancies	No. of Cycles	FR ^a^	95% CI ^a^	FR ^b^	95% CI ^b^
Total seafood, g/week												
<50	201	1070	1.00	Ref	1.00	Ref	1425	9847	1.00	Ref	1.00	Ref
50–99	356	1844	1.02	0.87–1.20	0.97	0.82–1.14	830	5014	1.11	1.03–1.20	1.06	0.98–1.14
100–199	745	3980	1.00	0.87–1.15	0.94	0.81–1.08	688	4125	1.11	1.02–1.21	1.05	0.96–1.14
≥200	516	2715	1.03	0.88–1.20	0.94	0.79–1.10	324	2090	1.06	0.95–1.19	1.00	0.90–1.13
Total unfried seafood, g/week ^c^												
<20							978	6839	1.00	Ref	1.00	Ref
20–49							676	4440	1.05	0.96–1.15	1.04	0.95–1.13
50–99							748	4511	1.12	1.02–1.22	1.05	0.96–1.15
100–199							591	3557	1.11	1.01–1.22	1.04	0.95–1.15
≥200							274	1729	1.09	0.96–1.23	1.03	0.91–1.17
Total fried seafood, g/week ^c^												
None							1726	11,079	1.00	Ref	1.00	Ref
1–4							593	3615	1.01	0.93–1.10	1.02	0.94–1.11
5–19							676	4573	0.94	0.87–1.02	0.97	0.90–1.05
≥20							272	1809	0.97	0.86–1.09	0.97	0.86–1.09
Shellfish, g/week												
None	107	655	1.00	Ref	1.00	Ref	1388	9347	1.00	Ref	1.00	Ref
1–9	880	4472	1.19	0.99–1.43	1.22	1.01–1.46	724	4471	1.06	0.98–1.15	1.01	0.93–1.10
10–49	768	4181	1.10	0.91–1.33	1.16	0.96–1.41	890	5501	1.07	0.99–1.16	1.02	0.94–1.10
≥50	63	301	1.23	0.93–1.63	1.21	0.91–1.61	265	1757	1.00	0.89–1.13	0.96	0.85–1.09
Unfried shellfish, g/week ^c^												
None							1494	10,052	1.00	Ref	1.00	Ref
1–9							881	5449	1.06	0.98–1.15	1.01	0.94–1.09
10–19							374	2246	1.10	0.99–1.22	1.05	0.95–1.16
≥20							518	3329	1.02	0.93–1.12	0.98	0.89–1.07
Fried shellfish, g/week ^c^												
None							2306	14,764	1.00	Ref	1.00	Ref
1–4							558	3548	0.98	0.91–1.07	0.98	0.90–1.06
5–9							341	2245	0.98	0.88–1.09	1.00	0.90–1.11
≥10							62	519	0.79	0.62–1.00	0.77	0.61–0.98
Fish oil supplementation												
No	1480	7746	1.00	Ref	1.00	Ref	2622	17,134	1.00	Ref	1.00	Ref
Yes	338	1863	0.97	0.87–1.08	0.94	0.84–1.06	645	3942	1.04	0.96–1.13	0.98	0.91–1.06

Ref = reference group. ^a^ Adjusted for energy intake (kcal/day). ^b^ Additionally adjusted for age, female BMI, physical activity, smoking status, alcohol intake, last form of contraception, intercourse frequency, parity, use of methods to improve pregnancy chances, daily multivitamin use, use of fish oil supplements, education, income, marital status, race/ethnicity (PRESTO only), healthy diet indices, and sugar-sweetened beverage intake. ^c^ Variables not available in Snart Foraledre.

**Table 3 nutrients-12-02276-t003:** Associations between dietary intake of marine-sourced long-chain omega-3 fatty acids and fecundability.

	Snart Foraeldre							PRESTO						
	g/week	No. of Pregs	No. of Cycles	FR ^a^	95% CI ^a^	FR ^b^	95% CI ^b^	g/week	No. of Pregs	No. of Cycles	FR ^a^	95% CI ^a^	FR ^b^	95% CI ^b^
Total marine fatty acids, percentile														
<25th	<1.00	448	2358	1.00	Ref	1.00	Ref	<0.37	775	5293	1.00	Ref	1.00	Ref
25th–49th	1.00–1.83	458	2484	0.96	0.86–1.08	0.93	0.83–1.04	0.37–0.60	779	5595	0.96	0.87–1.05	0.93	0.85–1.02
50th–74th	1.84–2.91	469	2411	1.02	0.91–1.15	0.98	0.87–1.10	0.61–1.03	845	5116	1.08	0.99–1.18	1.02	0.93–1.11
75th–89th	2.92–4.30	266	1434	0.97	0.84–1.11	0.92	0.80–1.06	1.04–1.64	534	2977	1.16	1.05–1.28	1.07	0.97–1.19
≥90th	≥4.31	177	922	1.03	0.88–1.21	1.00	0.85–1.18	≥1.65	334	2095	1.04	0.93–1.17	0.97	0.86–1.09
EPA percentile														
<25th	<0.29	448	2330	1.00	Ref	1.00	Ref	<0.09	752	5401	1.00	Ref	1.00	Ref
25th–49th	0.29–0.52	455	2540	0.94	0.83–1.05	0.90	0.80–1.02	0.09–0.16	806	5453	1.05	0.96–1.15	1.03	0.94–1.13
50th–74th	0.53–0.82	467	2413	1.01	0.90–1.14	0.97	0.86–1.09	0.17–0.31	846	5195	1.12	1.03–1.23	1.07	0.98–1.17
75th–89th	0.83–1.19	267	1441	0.95	0.83–1.09	0.91	0.79–1.05	0.32–0.54	537	2923	1.22	1.11–1.35	1.13	1.02–1.26
≥90th	≥1.20	181	885	1.09	0.93–1.27	1.06	0.90–1.25	≥0.55	326	2104	1.08	0.96–1.21	1.00	0.89–1.13
DPA percentile														
<25th	<0.07	457	2376	1.00	Ref	1.00	Ref	<0.07	776	5407	1.00	Ref	1.00	Ref
25th–49th	0.07–1.13	446	2484	0.95	0.84–1.06	0.92	0.82–1.04	0.07	838	5226	1.08	0.98–1.17	1.08	0.98–1.19
50th–74th	1.14–0.21	468	2389	1.03	0.92–1.15	0.99	0.88–1.11	0.08–0.11	797	5328	1.02	0.93–1.12	1.05	0.93–1.18
75th–89th	0.22–0.33	266	1444	0.96	0.84–1.11	0.92	0.80–1.06	0.12–0.15	512	3101	1.10	0.99–1.21	1.07	0.97–1.19
≥90th	≥0.34	181	916	1.05	0.90–1.23	1.04	0.89–1.22	≥0.16	344	2014	1.13	1.01–1.27	1.08	0.96–1.21
DHA percentile														
<25th	<0.64	450	2353	1.00	Ref	1.00	Ref	<0.20	769	5306	1.00	Ref	1.00	Ref
25th–49th	0.64–1.17	458	2478	0.96	0.85–1.07	0.92	0.82–1.04	0.20–0.36	784	5570	0.96	0.88–1.06	0.94	0.86–1.03
50th–74th	1.18–1.86	468	2405	1.02	0.91–1.14	0.97	0.86–1.09	0.37–0.61	840	5159	1.08	0.99–1.18	1.01	0.92–1.11
75th–89th	1.87–2.78	264	1454	0.95	0.82–1.09	0.90	0.78–1.04	0.62–0.96	535	2947	1.18	1.07–1.30	1.09	0.98–1.21
≥90th	≥2.79	178	919	1.04	0.89–1.21	1.01	0.86–1.18	≥0.97	339	2094	1.07	0.95–1.20	1.00	0.89–1.12

Ref = reference group. Total marine fatty acids = EPA + DHA + DPA. EPA = eicosapentaenoic acid, DHA = docosahexaenoic acid, DPA = docosapentaenoic acid. ^a^ Adjusted for energy intake (kcal/day). ^b^ Additionally adjusted for age, female BMI, smoking status, alcohol intake, physical activity, last form of contraception, intercourse frequency, parity, use of methods to improve pregnancy chances, daily multivitamin use, use of fish oil supplements, education, income, marital status, race/ethnicity (PRESTO only), healthy diet indices, and sugar-sweetened beverage intake.

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
