# Peer review of "The Association between Seafood Intake and Fecundability: Analysis from Two Prospective Studies"

_nutrients, 2020, doi:10.3390/nu12082276_

Round 1

Reviewer 1 Report

Elucidating the association of nutrients with reproductive performance is highly desired. Currently, there is conflicting evidence regarding the effects of seafood intake on reproductive parameters. The current manuscript describes the use of two prospective cohort studies for evaluation of fecundity in relation to seafood intake and marine-sourced omega-3 fatty acids. There was little association between intake of seafood or marine fatty acids and fecundability. However, a greater intake of fried shellfish was associated with reduced fecundability in North American participants.

The described studies provide evidence that contribute to reduce the current controversy in the field.

Author Response

We thank the reviewer for the kind comments. We can confirm that the first author's first language is English and that the English in this manuscript is of the highest quality possible. Thank you.

Reviewer 2 Report

nutrients-859667

I read with interest the manuscript titled “Seafood intake and fecundability in two prospective 2 cohort studies”, which falls within the aim of Nutrients. Authors analyzed the role of seafood on fecundability and found a decreased fecundability in patients who eat more fried shellfish.

The topic is really interesting, the Material and Methods section and the Discussion are well written but authors should clarify some points in the text as reported below.

Authors should consider the following recommendations:

- Title: Authors should consider modifying the title. An option could be “The impact of seafood intake on fecundability: analysis from two prospective studies”.

- Manuscript should be further revised by a native English speaker in order to correct several typos.

- Introduction: Authors analyzed the role of organic pollutants such as heavy metals and their impact on fecundity. This aspect should be improved, referring also to studies such as: De  Franciscis  P,  Guadagno  M,  Miraglia  N,  D’Eufemia  D,  Schiattarella  A,  Labriola  D,  et  al.  Follicular PB levels in women attending in vitro fertilization:    role  of  endometriosis  on  the  outcome.    Ital  J  Gynaecol  Obstet  2018;  30:  21–27. doi: 10.14660/2385-0868-92;

Schiattarella A, Colacurci N, Morlando M, Ammaturo F Pietro, Genovese G, et al. Plasma and urinary levels of lead and cadmium in patients with endometriosis. Ital J Gynaecol Obstet 2018; 30:47–52. doi: 10.14660/2385-0868-84

- Materials and Methods: was this study designed according to the STROBE Statement (PMID: 18064739), available through the EQUATOR Network (http://www.equatornetwork.org/)? It would be mandatory to report this information.

Author Response

We thank the reviewer for these helpful comments. We have the following responses to these comments:

Reviewer comment: Title: Authors should consider modifying the title. An option could be “The impact of seafood intake on fecundability: analysis from two prospective studies”.

Authors' response: We would prefer to retain the original title; however, if the editor and reviewers prefer, we would suggest the following alternative: "The association between seafood intake and fecundability: analysis from two prospective studies." Of the two newly-suggested titles, we prefer the one that phrases the title in non-causal terms. We felt that "impact" was too strong a word and we would prefer to use the term "association." We hope the reviewer agrees. Thank you.

Reviewers' comment: Manuscript should be further revised by a native English speaker in order to correct several typos.

Authors' response: the manuscript was indeed written by a native English speaker so we take this comment very seriously. We have read through the manuscript several times and have fixed all typos that we observed. We apologize for any errors in the original submission that may have given the impression that it was not written by a native English speaker. Some words are indeed unusual, such as "plaice" -- which is the actual name of a type of fish.

Reviewer's comment:  Introduction: Authors analyzed the role of organic pollutants such as heavy metals and their impact on fecundity. This aspect should be improved, referring also to studies such as: De  Franciscis  P,  Guadagno  M,  Miraglia  N,  D’Eufemia  D,  Schiattarella  A,  Labriola  D,  et  al.  Follicular PB levels in women attending in vitro fertilization:    role  of  endometriosis  on  the  outcome.   Ital  J  Gynaecol  Obstet  2018;  30:  21–27. doi: 10.14660/2385-0868-92; Schiattarella A, Colacurci N, Morlando M, Ammaturo F Pietro, Genovese G, et al. Plasma and urinary levels of lead and cadmium in patients with endometriosis. Ital J Gynaecol Obstet 2018; 30:47–52. doi: 10.14660/2385-0868-84.

Authors' response: We agree that these studies contribute to the literature on evaluating heavy metal intake and endometriosis, but neither of the above papers are studies that have directly evaluated seafood intake and fertility/fecundability, which was the outcome variable that we studied in our paper. Thus, we do not believe that they are appropriate for our literature review. Should the reviewers and editors feel strongly that these citations be added where we discuss heavy metals, we would be happy to comply. We would suggest that these associations be added at the end of the last sentence of the first paragraph (line 50, after citation 15). We would also want to add the following citation, which directly estimates fecundability (the outcome under study in this manuscript): Buck Louis GM, Sundaram R, Schisterman EF, Sweeney AM, Lynch CD, Gore-Langton RE, Chen Z, Kim S, Caldwell KL, Barr DB. Heavy metals and couple fecundity, the LIFE Study. Chemosphere. 2012 Jun;87(11):1201-7. PMID: 22309709. Thank you.

Reviewer's comment: Materials and Methods: was this study designed according to the STROBE Statement (PMID: 18064739), available through the EQUATOR Network (http://www.equatornetwork.org/)? It would be mandatory to report this information.

Thank you for this comment. We have added the following sentence to the revised paper on lines 100-101. "Both studies were designed according to the STROBE Statement (https://strobe-statement.org)." Please note that the link provided above by the reviewer is not correct.

Thank you again for all these excellent comments! We are very pleased to have the opportunity to revise and resubmit our work for future consideration.

Sincerely,

Dr. Wise and colleagues

Round 2

Reviewer 2 Report

Dear Authors

thank you for your modifies. Regarding the title the option "The association between seafood intake and fecundability: analysis from two prospective studies" suits well with the rest of the manuscript. Please change the title with this one.

Moreover, I suggest to add all references regarding the role of heavy metals as required and also the citation you referred (PMID: 22309709).

Kind regards

Author Response

Thank you for the opportunity to revise our paper. We provide a point-by-point response to the reviewers' comments below.

Reviewer's comment: Thank you for your modifies. Regarding the title the option "The association between seafood intake and fecundability: analysis from two prospective studies" suits well with the rest of the manuscript. Please change the title with this one.

Authors' response: We have changed the title as suggested.

Reviewer's comment: Moreover, I suggest to add all references regarding the role of heavy metals as required and also the citation you referred (PMID: 22309709).

Authors' response: We have added the citations as suggested. Thank you.